# Thermostatistics, Information, Subjectivity: Why Is This Association So Disturbing?

**Didier Lairez** 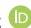

Laboratoire des Solides Irradiés, École Polytechnique, CEA, CNRS, IPP, 91128 Palaiseau, France;
didier.lairez@polytechnique.edu

**Abstract:** Although information theory resolves the inconsistencies (known in the form of famous enigmas) of the traditional approach of thermostatistics, its place in the corresponding literature is not what it deserves. This article supports the idea that this is mainly due to epistemological rather than scientific reasons: the subjectivity introduced into physics is perceived as a problem. Here is an attempt to expose and clarify where exactly this subjectivity lies: in the representation of reality and in probabilistic inference, two aspects that have been integrated into the practice of science for a long time and which should no longer frighten anyone but have become explicit with information theory.

**Keywords:** thermodynamics; statistical mechanics; information theory; subjectivity

## 1. Introduction

With great success, statistical mechanics provides *"the rational foundations of thermodynamics"* (Gibbs [1]), which thus becomes thermostatistics. However, *"[it] is notorious for conceptual problems to which it is difficult to give a convincing answer"* (Penrose [2]). Since their origin, these problems have been illustrated by famous enigmas that do not prevent the theory from advancing but are like pebbles in the shoe. I think about the two Gibbs' paradoxes (related to the mixing of two volumes of gas), the Poincaré–Zermelo paradox (related to the recurrence of dynamical systems), Loschmidt's paradox (related to the reversibility of the equations of mechanics) and its demonic version, the Maxwell's demon (a ratchet–pawl mechanism at the scale of particles). These enigmas all have one thing in common: they are all concerned with the second law of thermodynamics and entropy, the concept that was invented by Clausius [3] to account for the irreversibility of energy exchanges, linked by Boltzmann [4], Planck [5] and Gibbs [1] to probabilities and finally "enlightened" by Shannon's information theory [6].

"Enlightened" is wishful thinking because, although information theory resolves the inconsistencies raised by these enigmas (as will be shown), this contribution is far from being unanimously recognized. In recent textbooks, apart from one exception [7], information theory is either just mentioned but not really used [8] or totally ignored [9–12]. The situation is also ambiguous in the recent research literature.

The contribution of information theory to thermostatistics is two-fold, and each part must be well identified. The first one is linked to the encoding significance of entropy [13] and the relation it gives between energy and the information needed to reproduce the system as it appears to our senses, that is to say, to make a representation of it. The second, which is no less fundamental, is related to the "maximum entropy principle", which legitimates an inductive probabilistic inference based on our partial knowledge of the system [14–16] to describe its state of equilibrium. It legitimates prior probabilities (the first meaning of probability seen as a degree of belief) as opposed to a posteriori probabilities on which a frequentist inference could be performed.

This dual contribution allows for very efficient shortcuts of thought and for resolving inconsistencies in the theory, such as those illustrated by the enigmas mentioned above. This is not new, and these advantages are sometimes recognized even by those who do not defend the viewpoint of information theory and virulently combat it. For instance, *"Although information theory is more comprehensive than is statistical mechanics, this very comprehensiveness gives rise to objectionable when it is applied in physics and chemistry"* (Denbigh and Denbigh [17] p. 117). So, if information theory is not more widely adopted, it is either because its benefits are poorly understood (which we will also try to remedy) or because they are fully understood but rejected for epistemological rather than scientific reasons. In fact, the stumbling block is that information theory is seen as introducing subjectivity into physics, which is classified as a "hard science" practiced with rigor and objectivity.

In phenomenological thermodynamics, any system is considered a sort of "black box" with inputs and outputs in the form of heat and work exchanged with the surroundings. Phenomenological thermodynamics does not care about what is happening at a microscopic level inside the black box. It only deals with these inputs and outputs at the macroscopic scale, which are actually the only measurable quantities. How can a measurable quantity be subjective? It appears to be nonsense. Let us first clarify this point.

The only definition of the entropy $S$ of a system in thermodynamics (Clausius entropy) is in fact that of its variation for a reversible transformation (i.e., sufficiently slow compared to all relaxation processes of the system). Only, in this case, it is given by the exact differential $dS = T^{-1} dQ$ (where $T$ is the temperature and $Q$ the quantity of heat exchanged). But transformations, say from $A$ to $B$, are not reversible in most cases, so the corresponding variation in entropy cannot be measured. The only way to measure it is to close a thermodynamic cycle by returning to the initial state $A$, this time through a reversible transformation. The subjectivity of entropy lies precisely here: How to be certain that the cycle is indeed closed? This clearly depends on our knowledge of the initial state, which depends on the information we have on it: *"The idea of dissipation of energy depends on the extent of our knowledge"* (Maxwell [18]). "Objective" means that the quantity under consideration only depends on the object and not on the observer (the subject). In this paper, "subjective" means that the subject (the observer) plays a role. But this role is independent of the person of the subject; two scientists with the same information would reach the same conclusion [15]. Although this notion of subjectivity thus understood was already present in classical phenomenological thermodynamics, it completely disappeared with the advent of statistical mechanics.

Probably everything (and its opposite) has already been written on this subject, so the main purpose of this article is to attempt clarification and remove any ambiguity. In the first part, we will see how exactly subjectivity is brought to thermostatistics via entropy by two means: the encoding of a representation and the probabilistic inference, which are linked to the two features already evoked. Also, the last point will be compared to the alternative frequentist (objectivist) inference, namely the ergodic hypothesis.

The second part addresses the above puzzles and highlights the inconsistencies they raise in relation to the "objectivist" position. These inconsistencies are all removed with information theory (namely the "subjectivist" position) in a concise manner.

The last part tries to put the debate at an epistemological level: the objective versus subjective conceptions of entropy. It aims to extricate things, to show precisely where the arbitrariness lies and to answer the question: why is subjective entropy such a disturbing concept? A particular focus will be made on the filiation of the ideas behind the approach of information theory once applied to thermostatistics. This filiation ultimately corresponds to a particular conception of what science is, which originates from Plato's allegory of the cave and develops where modern representationalism (or indirect realism), empiricism, falsificationism and Bayesianism meet. That being exposed, everyone can decide whether this conception is natural or worrying, weigh it against the advantages provided by information theory and make a choice.

## 2. Information

### 2.1. Encoding, Information, Uncertainty

In everyday life, the question "How much information does this newspaper contain?" is understood in terms of the novelty of the meaning (the substance). With the advent of communication and computer sciences, an alternative signification concerns the minimum quantity of bits that would be needed to transmit, store and reproduce it later (the form). But the form is the physical support of the substance, and the novelty may lie in the form. Also, an entirely predictable source of "information" does not require storing data to be reproduced. So, both acceptations of the term "quantity of information" are linked. However, the latter has the advantage of being much more manageable. This was the approach of Shannon: *"[The] semantic aspects of communication are irrelevant to the engineering problem. The significant aspect is that the actual message is one selected from a set of possible messages. [Transmission and storage devices] must be designed to operate for each possible selection, not just the one which will actually be chosen since this is unknown at the time of design."* [6]. With Shannon, the message becomes a random variable to be lossless encoded and stored.

Consider a source (a thermodynamic system) that sequentially emits a random message $\mathbf{x} \in \Gamma$ (e.g., adopts a given microstate, $\Gamma$ is the phase space) according to a fixed probability distribution $p(\mathbf{x})$ (the system is at equilibrium). We plan to perform a lossless recording of the sequence to reproduce it exactly, for instance, to study and describe it later. Whatever the nature of the random events, if the number $W$ of their possibilities (the cardinality of $\Gamma$ or its "volume") is finite, we can establish a one-to-one correspondence table $m$ (a mapping) that assigns to each event $\mathbf{x}$ an integer $n = m(\mathbf{x})$ ranging from 1 to $W$:

$$m: \quad \mathbf{x} \in \Gamma \quad \longmapsto \quad n \in \{1, \dots W\} \tag{1}$$

So, recording the system's behavior (the source emission) would start by recording the correspondence table (the metadata), then continue by recording the sequential outcomes of the random integer variable $n$ (the data). This passes through the encoding of the latter, say a binary representation. Thus the question arises: what minimum number $H$ of bits per outcome should be provided for storage or transmission with a given bandwidth? By quantity of information emitted by the source, we mean this minimum number of bits per outcome for lossless recording.

#### 2.1.1. Fixed-Length Encoding

The central point is to seek the most economical encoding rule [19]. The first answer is to plan a fixed length per outcome (per word). Of course, for the decoder to be able to discriminate the end of a word from the start of the next, this conventional fixed length must be recorded with the metadata. The length must be large enough to store the largest integer $W$ outcome that is expected. Since $W = 2^{\log_2(W)}$, up to a rounding error,

$$H = \log_2(W) \tag{2}$$

The greater the number $W$ of possibilities, the greater the uncertainty of a given outcome and the greater the minimum length. This last equation allows us to consider $H$ either as the a posteriori average number of bits per outcome or as the a priori expected number of bits that should be scheduled to record upcoming events in case we have absolutely no idea about their actual probability distribution. In the former case, $H$ is a measure of the quantity of information that has been emitted by the source, whereas in the latter case, it is for the observer a measure of the uncertainty about the outcomes. These two facets are found in the usual meaning of probability.

#### 2.1.2. Variable-Length Encoding

Equation (2) is not an optimal solution for storing data because small numbers that only require a few bits take up the same storage space as large numbers. Variable-length encoding that uses just enough space for each outcome, i.e., $\log_2(m(\mathbf{x}))$ bits for outcome $\mathbf{x}$,

is better. Of course, this supposes a special encoding type, named prefix code, making it possible for decoding to identify the end of a given word and the beginning of the next. For instance, this can be performed using a delimiter. This also supposes that the encoding rules are recorded with the rest of the metadata, the size of which, however, will be assumed to be negligible compared to that of the data (which is legitimate for a long sequence of recordings). With variable-length encoding, the average number of bits per outcome is

$$H = \sum_{\mathbf{x} \in \Gamma} p(\mathbf{x}) \log_2(m(\mathbf{x})) \tag{3}$$

where $p(\mathbf{x})$ is the probability of the outcome $\mathbf{x}$ to which was assigned the integer $m(\mathbf{x})$, according to the mapping.

Let us first examine the special case where $p(\mathbf{x})$ takes the constant value $1/\mathcal{W}$ whatever $\mathbf{x}$. Equation (3) gives

$$H = \frac{1}{\mathcal{W}} \sum_{\mathbf{x} \in \Gamma} \log_2(m(\mathbf{x})) \tag{4}$$

The smallest values for the series $n = m(\mathbf{x})$ are obtained by starting from $n_1 = 1$ and then applying the rule $n_i = n_{i-1} + 1$. This sequence is that of natural numbers up to $\mathcal{W}$, so one obtains

$$H = \frac{1}{\mathcal{W}} \sum_{i=1}^{\mathcal{W}} \log_2(i) = \frac{1}{\mathcal{W}} \log_2(\mathcal{W}!) \tag{5}$$

For large $\mathcal{W}$, the Stirling formula leads to

$$H = \log_2(\mathcal{W}) + o(1) \tag{6}$$

which is asymptotically the same as Equation (2). Note that the same result is obtained whether no a priori information is known about the outcomes except that it is bounded (Equation (2)) or if we know in advance that outcomes obey a uniform probability distribution (Equation (6)).

Variable-length encoding is not economic of storage space for a uniform probability distribution. But there remain others. For non-uniform distributions, it is possible to choose $m(\mathbf{x})$ as being an increasing function of improbability $1/p(\mathbf{x})$, so that the values that require the most storage space are mapped to the rarest events. Different rules of assignment can be applied. For instance, according to the median, one can split $\Gamma$ into two subsets labeled 0 and 1. The first encoding bit for $m(\mathbf{x})$ is the label to which subset $\mathbf{x}$ belongs, and the other bits are obtained by subsequent similar recursive dichotomies. This procedure, named Fano encoding [20], gives a near-optimal encoding length. Shannon [6] showed that in no case can the average length per outcome be less than

$$H = \sum_{\mathbf{x} \in \Gamma} p(\mathbf{x}) \log_2(1/p(\mathbf{x})) \tag{7}$$

Note that the uniform distribution is a special case of this last equation.

### 2.1.3. Information Encoding and Energy

To factor $\ln 2$, one recognizes in Equations (2) and (6) the formula for the Boltzmann entropy of an isolated system (microcanonical), and in Equation (7), that of Gibbs entropy of a closed system (canonical). In both cases, one can write

$$S = H \ln 2 \tag{8}$$

which is called Shannon entropy [21] (in this paper, the temperature is in Joule, so the entropy is dimensionless). The Boltzmann–Gibbs entropies are in reality special cases of that of Shannon, for which the random events would be the different microstates that a thermodynamic system can adopt.

Here, let us recall that the formula for the Gibbs entropy was obtained from the canonical distribution of energy levels and from an identification with certain thermodynamic equalities involving Clausius entropy [22]. The derivation of the Gibbs entropy formula is entirely dependent on thermodynamics. The Shannon entropy, for its part, is obtained independently of thermodynamics but with the aim of optimizing the encoding size, an aim that is reminiscent of the idea that Gibbs entropy is at its maximum at thermodynamic equilibrium. The equality of the formula and the idea behind it are likely not coincidental.

The Shannon entropy of the distribution of microstates and the Boltzmann–Gibbs entropy are the same quantity. As the latter is the same as the Clausius entropy, the three are one and the same quantity. Hence, there is a connection between energy on one side and information/uncertainty on the other.

To be more precise about this connection, let us recall some thermodynamics. Consider a system with internal energy $U$. For any quasistatic process it undergoes, one can write

$$\Delta U = Q + W, \tag{9}$$

where $Q$ is the heat exchanged with the surroundings and $W$ is the work defined as the complementary part of $Q$ in virtue of the conservation law (first law of thermodynamics).

The second law of thermodynamics is two-fold: the first part defines the entropy $S$ as a state quantity linked to the heat exchanged for a reversible process, for which the entropy is given by its exact differential $dS = dQ/T$. Whereas the second is the Clausius inequality, which concerns the general irreversible case. In this paper, for the sake of simplicity, we will consider only processes at a constant temperature, allowing us to more easily integrate $\int T^{-1} dQ$. Then, the two parts of the second law become

**Second law of thermodynamics (§1)**:
*There exists a state quantity S, whose variation for a reversible process is such as $Q = T\Delta S$, where T is the temperature.*

**Second law of thermodynamics (§2.1)**:
*Clausius inequality: in all cases,*

$$Q \leq T\Delta S \tag{10}$$

The energy exchanges can be seen as a dissipation ($Q$) of an energy cost ($W$) because, generally, heat is unwanted and work is more valued. So, at constant internal energy ($\Delta U = 0$), say at a constant temperature for a gas, the Clausius inequality tells us that the energy cost to achieve a process is always greater than $-T\Delta S$ (i.e., $W \geq -T\Delta S$).

The twin of the Clausius inequality in terms of the quantity of information emitted by the source, or equivalently, in terms of the uncertainty about its emission, is obtained with Equation (8), leading to $W \geq -T\Delta H \ln 2$. The second part of the second law (§2) can thus be rewritten as

**Second law of thermodynamics (§2.2)**:
*The energy cost W to vary the uncertainty about the microstate of a system by $\Delta H$ is always*

$$W \geq -T\Delta H \ln 2 \tag{11}$$

The acquisition of data about the system via measurement of certain properties (for instance, the boundary of the phase space) is a way to reduce the uncertainty about the outcomes. So, the previous equation can be expressed per bit ($\Delta H = -1$) of acquired data:

$$W_{\text{acq/bit}} \geq T \ln 2 \tag{12}$$

Equations (11) and (12) are the key equations for linking energy and information. There is nothing more than that.

The two above versions of the second law (§2.1 and §2.2) express exactly the same thing but in different ways. The first speaks of heat dissipation and entropy, whereas the second prefers to speak of work and uncertainty. It is thus legitimate to question the real usefulness of the notion of information encoding in thermostatistics. The first answer is that a link between two fields of knowledge is part of what we call understanding something. The second answer is that the link between information encoding and energy allows us to express certain ideas in a more concise and consistent way. In particular, it provides shortcuts to solving thermostatistic enigmas. So, if a theory is ultimately an economy of thought [23–25], this is undoubtedly progress. The third answer is that entropy, so defined as an uncertainty, forms a package with the probabilistic induction that will be discussed in the following section.

### 2.1.4. Stability of Equilibrium

The second law of thermodynamics is often expressed for an isolated system to which neither heat nor work is exchanged with the surroundings ($Q = W = 0$). So, Equation (10) becomes

$$T\Delta S \geq 0 \tag{13}$$

This leads to another version of the second law:

**Second law of thermodynamics (§2.3):**
*The entropy of an isolated system cannot spontaneously decrease.*

Or, in terms of information,

**Second law of thermodynamics (§2.4):**
*The uncertainty (the quantity of information) about the microstate of an isolated system cannot spontaneously decrease.*

Classical and phenomenological thermodynamics is traditionally only concerned with equilibrium stricto sensu: *"a system is in an equilibrium state if its properties are consistently described by thermodynamic theory"* (H.B. Callen [26] p. 15). In classical thermodynamics, equilibrium is by definition a stationary stable state and the "equilibrium state" is a pleonasm: no states other than those at equilibrium are defined (by state quantities).

With Boltzmann–Gibbs entropy, probabilities come into play for the description of the thermodynamic equilibrium. Therefore, an isolated system can now fluctuate and deviate slightly from equilibrium. As this notion was absent in phenomenological thermodynamics, we are now faced with this problem: How can the equilibrium be stable? What is the restoring force of the system when it deviates from equilibrium? To ensure the stability of equilibrium, if we do not want to postulate it, we need an additional ingredient in the form of an alternative postulate or a definition of the nature of equilibrium. This definition can be the following:

**Definition of equilibrium (v1):**
*The equilibrium of an isolated system is the state of maximum entropy.*

Or, equivalently,

**Definition of equilibrium (v2):**
*The equilibrium of an isolated system is the state of maximum uncertainty about its microstate.*

With these definitions, the restoring force comes from the second law: due to fluctuations, the system may deviate from equilibrium but will return to it spontaneously.

### 2.2. Inductive Probabilistic Inference

The first program of statistical mechanics is to calculate certain observable macroscopic quantities of a thermodynamic system at equilibrium from the average of certain random variables that are relevant at a microscopic level. For instance, calculating the

temperature from the average kinetic energy of particles. These averages are computed over a probability distribution.

The central point is thus to determine which random variable to consider and what probability distribution it is supposed to obey. That is to say, make a statistical inference.

### 2.2.1. Subjective versus Objective Probabilities

Two different types of statistical inferences are traditionally distinguished: "probabilistic inference" and "frequentist inference", which depends on how probabilities are defined.

1.  Subjective (or prior) probabilities are reasonable expectations or degrees of belief that one thing or another will happen. They are subjective in that they depend on our knowledge of the system.
2.  Objective relative frequencies of occurrence (over an ensemble) of one thing or another that actually happened (or a posteriori probabilities). They are supposed to be tangible property of the system.

Subjective probabilities are general and can always apply, so they are de facto the most common on which to base a decision. But their arbitrary nature poses a problem without a rational criterion to assign a value to them. They appear illegitimate and may turn out to be false a posteriori. In contrast, frequencies are reliable, provided that the corresponding measurement has been carried out. But this is often impossible or at least not possible before making a decision. Their use is conditioned on the existence of an ensemble or at least on the hypothesis of its existence, provided that it can be performed in a consistent manner.

Consider the game of die. The die is cubic and offers six possible outcomes. Also, from a symmetry argument, there is no reason to believe that one is more likely than another. Prior to any toss of a die, we can reasonably assign to the outcomes a uniform discrete probability distribution lying from 1 to 6. This reasoning, which accords with common sense [27], is called "Laplace's principle of insufficient reason" (or "principle of indifference" [28]). It is a typical example of probabilistic inference.

An interesting point is that the most reasonable decision for a bet would be exactly the same if we knew in advance that the die is loaded (we knew in advance that the distribution is not uniform) but did not know which number is favored. The first assignment of probabilities is ultimately based on a criterion that seems much more arbitrary ("there is no reason to believe otherwise") than waiting for a few tosses of the die and estimating the frequency distribution from the sampling of outcomes (make a frequentist inference). The decision is based on prior probabilities that do not seem legitimate but are the only ones available.

In the problem of information processing, faced with an unknown source that we want to record, we must begin by using one or other of the encoding rules and then eventually use an adaptive procedure to reconsider (to update) the encoding according to the observations. For a lossless recording, the best choice to begin with is a fixed-length encoding (with an overestimated number of possible outcomes if it is unknown). This choice maximizes the compatibility of the encoding procedure with future incoming data. In Section 2.1, we saw that this choice amounts to assigning a uniform prior distribution for the outcomes and is therefore in agreement with the principle of insufficient reason. We know in advance that this prior distribution is certainly not the true one, but this does not prevent it from being chosen rationally. This choice is the best we can make; any other would be judged to be irrational.

The problem of assigning a prior distribution is ultimately reduced to the search for an optimal compromise between two contradictory goals: (1) avoid any loss of information; (2) avoid an unnecessary volume of storage. This problem of optimization is formalized and generalized with another aspect of Shannon's information theory that essentially aims to optimize the use of our prior knowledge of the source for the statistical inference.

2.2.2. Maximum Entropy Probabilistic Inference

Suppose we are dealing with a source that emits outcomes $i$ about which we have only a partial knowledge of the true probability distribution $p_i$, so different distributions can potentially satisfy these constraints. To start recording the source, we need to assign a prior distribution that *"agrees with what is known, but expresses a 'maximum uncertainty' with respect to all other matters"* (Jaynes [15]), and thus leaves a maximum chance of compatibility with subsequent data. The problem is reduced to maximizing the uncertainty, which therefore remains to be measured.

In addition to the definition of entropy as the optimum length to encode the outcomes of a source, Shannon [6] introduces another idea. He asks, *"Can we find a measure [...] of how uncertain we are of the outcome?"* and continues, *"If there is such a measure, say $H(p_1, p_2, \ldots)$, it is reasonable to require of it the following properties:"* (1) being continuous in $p_i$; (2) increasing in $\mathcal{W} = 1/p_i$ for a uniform distribution; (3) being additive over different independent sources of uncertainty. Then, Shannon demonstrated that the only function (the only measure of uncertainty) that fulfills these requirements is to a factor $\sum p_i \ln(1/p_i)$, i.e., the Shannon entropy. Hence, the theorem

**Maximum entropy theorem**: *the best prior distribution $p(x)$ that maximizes the uncertainty on $x$ while being consistent with our knowledge is the one that maximizes the Shannon entropy.*

The validity of this theorem is ultimately determined by what is supposed to be required for a measure of uncertainty. These requirements play the role of starting axioms for their demonstration. As natural as they seem, some may find them arbitrary, wondering why not prefer others, leading to another function being maximized. Shore and Johnson [16] start with a completely different requirement, a consistency axiom that can be stated like this: *"if the problem of assigning a prior distribution can be solved in more than one way of taking the same information into account (for instance in different coordinate systems), the results should be consistent"*. On which everyone should agree. Then, Shore and Johnson [16] prove that the only procedure satisfying this requirement of uniqueness is that of maximizing Shannon entropy. Given a random variable **x**, the maximum entropy theorem provides a legitimate method to determine the prior probability distribution $p(\mathbf{x})$.

However, to complete the program of statistical mechanics remains the first point raised at the beginning of this section: which random variable **x** to consider. For instance, imagine a random variable $x \in [0, \pi]$ with a uniform distribution, $\sin(x)$ is also a random variable whose distribution is not uniform but has a maximum for $x = \pi/2$. Thus, using the maximum entropy criterion directly on $x$ or on $\sin(x)$ leads to different distributions *in fine* for $x$. However, *"among all these distributions there is one particular one, corresponding to the absolute maximum of entropy, which represents absolutely stable equilibrium."* (Planck [5] p. 32). Jaynes [29] outlines a crucial point. The assumption of uniqueness of equilibrium actually automatically brings to our knowledge other crucial information: the solution is not supposed to depend on the orientation of the observer (invariance under rotation), on its position (invariance under translation), or on the scale it is considered (invariance under scaling). Among all the possible variables describing a system, considering only those whose distributions are invariant in form under similarities avoids inconsistent results (leads to the same result). Hence, the definition of equilibrium:

**Definition of equilibrium (v3)**:
*The equilibrium of a system is the only state that maximizes the uncertainty on variables whose distributions are similarity-invariant in form.*

This definition plus the maximum entropy theorem can be expressed all-in-one in the form of the so-called maximum entropy principle, which is actually also an alternate definition of the equilibrium:

**Definition of equilibrium (v4)**:
*The equilibrium of a system is the only state that maximizes the Shannon entropy of variables whose distributions are similarity-invariant in form.*

Two classical examples of similarity-invariant distributions mentioned in the above definitions are those of Boltzmann and Gibbs:

- Boltzmann [4] considers the phase space of one single particle ($\Gamma_1 \subset \mathbb{R}^6$). For $N$ particles, provided that their phase spaces are identical, the probability distribution $p(\mathbf{x})$ he considers is that of finding one particle in a given elementary "volume" $\mathbf{x}$ ($p(\mathbf{x})$ is the particle density). The H-function he defines is the Shannons's entropy of $p(\mathbf{x})$.
- Gibbs [1] considers directly the phase space of $N$ particles ($\Gamma \subset \mathbb{R}^{6N}$). $p(\mathbf{x})$ is the probability for the system to be in microstate $\mathbf{x}$. The corresponding Shannon's entropy is the same as that defined by Gibbs, who showed that it is also equal to the Clausius entropy of the system.

Maximizing the entropy of either distribution yields two consistent descriptions of equilibrium. In practice, the maximization procedure is a variational calculus taking into account the constraints imposed by our knowledge about the system [6]. For instance, suppose that the only thing we know about $p(\mathbf{x})$ is that it has a finite support (the minimum required for a discrete distribution to be properly normalized), then the best distribution is uniform (this is the microcanonical distribution for an isolated system). If $p(\mathbf{x})$ is only known to have a positive support, allowing $\mathbf{x}$ to have a finite average value, then the best distribution is the exponential decay (this is the canonical distribution of energy levels for a closed thermalized system).

In these last two definitions of equilibrium (v3 and v4), the statement of uniqueness should not be controversial. The core of the controversy lies in the rest. One may consider that there is absolutely no reason why the system would actually maximize the uncertainty we, as observers, have about its microstate. But making another inference would be neither optimal nor reasonable. Moreover, the similitude between the first definitions of equilibrium (v1 and v2) that were given in Section 2.1.4 and these last two is noticeable. Whereas the first definitions were based on thermodynamical considerations and the need for the theory to define the equilibrium as stable, the second are more general (not only concerned with microstates) and emerge from a reasoning totally free from thermodynamics and any considerations regarding stability. This reasoning is an inductive probabilistic inference, which will be called "subjective", in contrast with the alternative, called "frequentist", which is believed to be more objective.

### 2.2.3. Alternative "Frequentist" Inference

Statistical mechanics did not wait for information theory to infer distributions at equilibrium. Alternative approaches focus on the distribution of microstates. In addition, the problem lies uniquely in deciding what is the distribution for an isolated system. Because, this being determined, the distribution for any closed subpart can be deduced [22]. These alternative approaches are essentially of two kinds.

The first is based on the already-mentioned principle of insufficient reason, which is renamed for the circumstance "fundamental postulate of statistical mechanics" [30]. In fact, it is nothing other than a less formalized and less general expression of the maximum entropy principle [31].

We will only focus on the second alternative approach, that of "frequentists", which intends to adopt an objective point of view. To compensate for the lack of knowledge of the system, the idea is to make a strong hypothesis, that of "ergodicity". This hypothesis essentially aims to fulfill the prerequisite for the definition of probabilities as relative frequencies: the existence of an ensemble on which to calculate them.

Consider an isolated thermodynamic system; its phase space $\Gamma$ is the set (ensemble) of all possible microstates $\mathbf{x}$ of probability $p(\mathbf{x})$ under which copies of the system can be found. Take one of these copies. It is dynamical, that is to say, over time it continuously undergoes a transformation $F : \Gamma \mapsto \Gamma$, allowing one to define, from the initial condition $\mathbf{x}_0$, a trajectory (an orbit) as the set of points in the phase space $\mathcal{T}(t) = \{\mathbf{x}_0, F(\mathbf{x}_0), F^2(\mathbf{x}_0), \ldots F^t(\mathbf{x}_0)\}$. The ergodic hypothesis is that this trajectory will finally pass recurrently through all points of

the phase space at the frequency $p$, which is therefore supposed to remain unchanged over time (the transformation $F$ preserves the volume of the phase space).

For our concern of determining which probability distribution is that of microstates, one consequence of the ergodic hypothesis is that "volume preserving" transformation means that for any point $\mathbf{x}$ of the trajectory,

$$p(\mathbf{x}) = p(F^{-1}(\mathbf{x})) \tag{14}$$

That is, the probability of a given consequent microstate is that of its antecedent. The probability distribution is therefore uniform and unchanged throughout the trajectory, which will ultimately cover the entire phase space. We thus obtain the microcanonical distribution we were looking for.

For comparison with the maximum entropy approach, it is interesting to express the hypothesis of ergodicity in the same form as previously, i.e., in the form of a definition of equilibrium:

**Definition of equilibrium (v5)**:
*The equilibrium of a system is the only state where the distribution of microstates is the same over its time transformation as over an ensemble of its copies.*

It immediately appears that this definition, contrary to the previous ones, does not contain any warranty of the stability of the equilibrium. There is no restoring force for the equilibrium, which, so defined, is not an attractor for the system. This point is one of the sources of the inconsistencies that will be discussed in the following.

At the basis of the hypothesis of ergodicity is the fact that deterministic Hamiltonian systems (according to Liouville's theorem) are volume-preserving and thus ergodic. In this context, the trajectory of the system in the phase space is a chain of causality. So, Equation (14) is more particularly interpreted as follows: the probability of a given consequent microstate is that of its "cause".

Still, for the sake of comparison, the detailed logical steps of the inductive reasoning for the ergodic hypothesis, which make it natural to us, are the following: (1) a set of atoms (a thermodynamic system) is analogous to a set of colliding rigid spheres (in the time of Maxwell and Boltzmann, as the notion of atom itself, this was far from obvious); (2) usually, a set of colliding rigid spheres comes under classical mechanics; (3) usually, the equations of motion of mechanics alone determine the future state of classic mechanical systems (this is generally the meaning of the word "deterministic", but there exist exceptions of non-deterministic classic mechanical systems, e.g., the "Norton's dome" [32]). From this is the following: (4) given a microstate $\mathbf{x}$ adopted by the system, the equations of motion of classical mechanics alone determine the future microstate $F(\mathbf{x})$ (Equation (14)).

In short, there is no definitive proof for the system to be ergodic, as measurements are not possible, but we consider this very likely. The reasoning is ultimately not that far from a probabilistic inference, but much less explicit on this point than that of maximum entropy.

## 3. Enigmas

Information theory introduces subjectivity in thermostatistics in two different and related manners: (1) the encoding of a representation of the system and its link with energy; (2) a probabilistic inference. In what follows, these two features are used to resolve the inconsistencies raised by the thermostatistic enigmas quoted in the introduction.

Here, these enigmas are classified into two categories, paradoxes and demons, which do not have the same level of importance. Paradoxes raise inconsistencies that cannot be removed by classical statistical mechanics without information theory. As for demons and the devices they drive, they do not actually introduce inconsistencies, in so far as they are physical systems that obey the second law of thermodynamics, as do all others. But they can rather be considered the first evidence of the link between information and energy that was given by Maxwell, Zermello and Szilard well before Shannon.

### 3.1. Gibbs Paradoxes

#### 3.1.1. The Problems

Consider a system $D$ (for disjoined) made of two adiabatic containers, denoted $A$ and $B$, of the same volume $V$, each containing the same quantity $N$ of an ideal gas (of the same species or not) with the same degree of freedom and at the same temperature $T$ (see Figure 1). So, the entropy of the two sub-systems is equal $S = S_A = S_B$. From the additivity of entropy,

$$S_D = 2S \tag{15}$$

Joining these two volumes by removing the partition between them results in another system $J$ (for joined). The process is never accompanied by any observable thermodynamic effect: neither heat nor work is exchanged with the surroundings. To determine whether or not this is accompanied by an increase in Clausius entropy, we must go back to the disjoined state (close the cycle) by a reversible process and measure heat exchange. Here, two paradoxes arise and continue to be debated for 150 years (for a review of the debate, see, e.g., the papers in [33,34]).

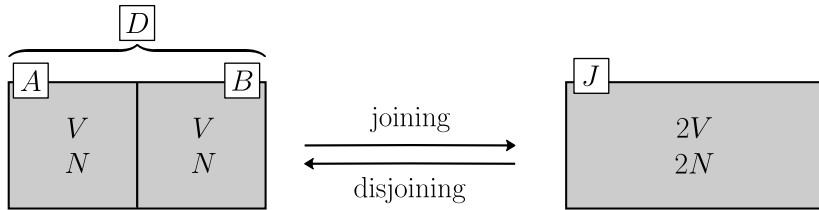

**Figure 1.** Usual image of the joining–disjoining thermodynamic cycle of two volumes of gas of the same species: removing the partition joins the two volumes and putting it back restores the system to its initial state with neither work nor heat exchanged with the surroundings.

If the two gases are identical, according to thermodynamic phenomenology, replacing the partition between the two containers allows the system to return to its initial state. This is performed without heat exchange, leading to the conclusion that the two states have the same Clausius entropy.

$$\Delta S = S_J - S_D = 0 \tag{16}$$

However, if initially the two gases differ, removing the partition mixes them, and just putting it back is insufficient to separate them again. The separation can be performed by two isothermal compressions against two pistons equipped with different semi-permeable membranes [35]. The first piston is only able to compress one species (from $2V$ to $V$) and the second only the other species (by the same ratio). The total work is thus equal to $W = -Q = T\Delta S = T \times 2N \ln 2$, leading to

$$\Delta S = S_J - S_D = 2N \ln 2 \tag{17}$$

In Equations (16) and (17), $\Delta S$ is called the entropy of mixing.

The first paradox mentioned by Gibbs [36] is that $\Delta S$ is a bivalued discontinuous step function of the dissimilarity of the gases, but it can legitimately be expected to be continuous like other property variations in the system (density, refractive index...).

The second paradox comes from statistical mechanics. The Boltzmann entropy of the disjoined state is $S_D = 2S = 2N \ln V$, whereas it is $S_J = 2N \ln 2V$ for the joined state. This leads to the difference $\Delta S = 2N \ln 2$. This is true regardless of the gas species, whether identical or not. If statistical mechanics solve the paradox of discontinuity, it raises another: Why do the Boltzmann and Clausius entropies differ when they should be the same?

#### 3.1.2. Usual Solutions

Concerning the first paradox, the consensus is that the discontinuity is not problematic. In fact, the dissimilarity of two species of atoms is discontinuous; thus, that of entropy is not

a problem. In terms of the classification of Quine [37], the discontinuity is mostly treated in the literature as a veridical paradox (the two premises are correct but not inconsistent; for a different approach, see [38]).

As for the second paradox, it is most of the time treated as a falsidical paradox (at least one premise is wrong). As phenomenology is the final arbiter, the calculation of the cardinality of the phase spaces must be reconsidered (corrected).

Justifications for this correction are mainly of two kinds. Denote $\mathcal{V}_D = \mathcal{V}^2$ and $\mathcal{V}_J$ as the cardinality of the phase spaces for systems $D$ and $J$, respectively, in the case where all the particles are different and clearly identified with a label, such as a serial number. Also, denote $\mathcal{W}_D = \mathcal{W}^2$ and $\mathcal{W}_J$ as the corresponding cardinality in the case where all particles are identical.

The first approach to justify a correction is based on the notion of the indistinguishability of particles, which comes from quantum mechanics [39]. For $N$ indistinguishable particles, their $N!$ permutations give the same microstate, which must therefore be counted only once, leading to $\mathcal{W} = \mathcal{V}/N!$. This is known as the correct Boltzmann counting, leading to

$$\mathcal{W}_D = \frac{\mathcal{V}_D}{N!N!}, \quad \mathcal{W}_J = \frac{\mathcal{V}_J}{(2N)!} \tag{18}$$

The second approach remains within the framework of classical mechanics, where particles (even identical) are always distinguishable in the sense that they always have distinct trajectories, allowing them to be (in principle) traceable and thus identified at any time. When partitioning the system into two compartments, particles can be combined in $(2N!)/N!N!$ different manners into the two separate compartments [40–42]. It follows that the number of possible results for the disjoined state is increased by this multiplicative factor, leading to

$$\mathcal{W}_D = \mathcal{V}_D \frac{(2N)!}{N!N!}, \quad \mathcal{W}_J = \mathcal{V}_J \tag{19}$$

It follows that, with the two approaches, the entropy of mixing two identical gases is the same,

$$\Delta S = \ln\left(\frac{\mathcal{W}_J}{\mathcal{W}_D}\right) = \ln\left(\frac{\mathcal{V}_J}{\mathcal{V}_D} \frac{N!N!}{(2N)!}\right) \tag{20}$$

but for different reasons [43]. In both cases, the second Gibbs paradox is claimed to be solved because, by using an approximation of the Stirling formula, one obtains $\ln(N!N!/(2N)!) = 2N\ln(N) - 2N\ln(2N) = -2N\ln 2$. So, the excess of entropy (Equation (17)) obtained with the Boltzmann equation is corrected.

It is important to outline that Equation (19) for $\mathcal{W}_D$ counts the number of all possible disjoined microstates. That is to say, $\mathcal{W}_D$ is the cardinality of the phase space viewed as an ensemble of different possibilities, including the different possible combinations in the repartition of particles. But once the partition is in place, a given disjoined microstate thus obtained will never by itself have a dynamic trajectory allowing it to reach another repartition (the repartition is frozen). In other words, the dynamics of a disjoined state cannot allow all the possibilities accounted for by Equation (19) to be explored. The disjoined state is no longer ergodic (this is noted in [44]). It follows that Equation (19) is implicitly valid if the corresponding entropy is an uncertainty about the actual state of the disjoined system. Probabilities are prior probabilities and not frequencies. In a classical mechanics framework, the above solution automatically places us implicitly in a "subjectivist" rather than in a "frequentist" position. That is to say, there is no solution to the second Gibbs paradox in the framework of classical statistical mechanics and frequentist (ergodic) inference. The solution is necessarily quantum or based on a probabilistic inference. It is up to the reader to decide which one is more consistent and natural. This often goes unnoticed.

The "subjectivist" position to solve the second Gibbs paradox is explicit in some papers, which are nevertheless largely in a minority (see, e.g., [45–48]). But even in these latter papers, the paradox of discontinuity is either eluded or treated as veridical.

### 3.1.3. Yet Another Solution

The aim of this section is to show how, from the information point of view, the dissimilarity of two gaseous contents is not bivalued but gradual, and Shannon's entropy is too, that is, it is as close as possible to a continuous function with atomistic matter. So, the paradox of discontinuity is actually falsidical. In doing so, the second paradox is also solved by considering it to be veridical.

Let us start by observing that thermodynamics considers cycles performed repeatedly and reproducibly. Therefore, if the two gases are identical, the representation of a joining/disjoining cycle does not care about the following:

1.  The exact number of particles in each compartment up to the standard deviation $\sqrt{N}$ of the binomial distribution;
2.  The traceability of particles (this information is only relevant if the two gases initially differ).

The correct image of the joining–disjoining cycle is given in Figure 2 (instead of Figure 1). It is to this very cycle that thermodynamics refers in Equation (16). The calculation of $\Delta S$ in terms of probabilities has thus to be performed by accounting for these two useless pieces of information. Accounting for the latter is very common and leads to the correct Boltzmann counting ($-\ln N!$ term); accounting for the former requires in addition the use of the exact Stirling formula $\ln(N!) = N\ln(N/e) + \ln(\sqrt{2\pi N}) + o(1)$ (rather than the usual approximation that consists in the first term only). It can be found in [49]. Here, a different derivation is proposed that avoids the term $-\ln(N!)$ and allows us in doing so to solve the paradox of discontinuity.

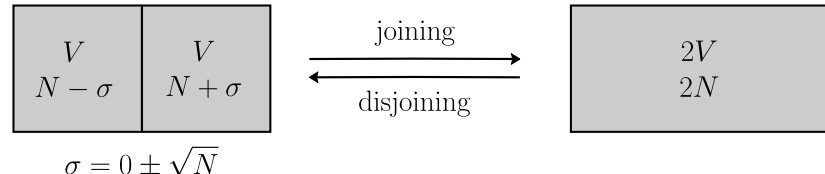

**Figure 2.** Correct image of a joining–disjoining thermodynamic cycle supposed to be repeatable (to be compared to Figure 1): putting back the partition leaves each compartment with the same number of particles up to the standard deviation $\sqrt{N}$.

Consider a first kind of cycle (see Figure 3, top) that consists of just moving the partition between the two compartments, denoted $A$ and $B$. Let $t_0$ be the time just before the first cycle starts. After the first cycle ends, all pieces of information about the exact contents of $A$ and $B$ at time $t_0$ are lost. At the end of each cycle in the disjoined state, the number of particles per compartment is always $N \pm \sqrt{N}$. Also, after $t_0$, any information about the traceability of particles is lost. So, the uncertainty concerning these two features, the exact number of particles and traceability, is unchanged by further cycles, and so is the Shannon entropy. Therefore, by considering these further cycles, the Shannon entropy of mixing is zero, just like the Clausius entropy:

$$\Delta S_1 = 0 \tag{21}$$

Imagine that we know for certain that the two compartments initially had exactly the same number of particles $N \pm 0$ and that we want to retrieve this information when restoring the disjoined state (see Figure 3, middle). The procedure for the gas partition can be the following:

1.  Put all particles in a separate box;
2.  Partition the empty volume $2V$ into $A$ and $B$;
3.  Iteratively take one pair $(a, b)$ of particles; put one particle (either $a$ or $b$) in compartment $A$ and the other in $B$.
4.  After $N$ iterations, the two compartments have exactly the same number of particles.

There are four possibilities to arrange $a$ and $b$ in two boxes: $\{.|ab, a|b, b|a, ab|.\}$, and only $a|b$ or $b|a$ are convenient. Therefore, each iteration divides the number of possibilities by two and gives us 1 bit of information. The $N$ iterations of the overall procedure and Equation (12) provide the corresponding entropy of mixing:

$$\Delta S_2 = N \ln 2 \tag{22}$$

Note that the procedure can be stopped at any iteration if we are satisfied by the uncertainty on $N$ would lead a random repartition of the rest of the particles. So, depending on our wish, the entropy of mixing can take any value from 0 to $N \ln 2$ by a step of $\ln 2$.

Imagine that we know for certain that at $t_0$, compartment $A$ was filled with isotope $a$ and compartment $B$ with isotope $b$ (see Figure 3, bottom). So, we are not satisfied by the previous procedure and want to restore the original state exactly. In other words, we want to preserve the traceability. To achieve this, among the two possibilities $\{a|b, b|a\}$ in the previous procedure, we must choose $a|b$. Here again, at each iteration, the number of possibilities is divided by two and gives us one additional bit of information. So, at the end, compared to the previous state, the entropy has decreased by an additional amount $N \ln 2$. Finally, if we consider traceability as crucial, the Shannon entropy of mixing is

$$\Delta S_3 = 2N \ln 2 \tag{23}$$

Here again, the procedure can be stopped at any iteration according to which degree of impurities is acceptable.

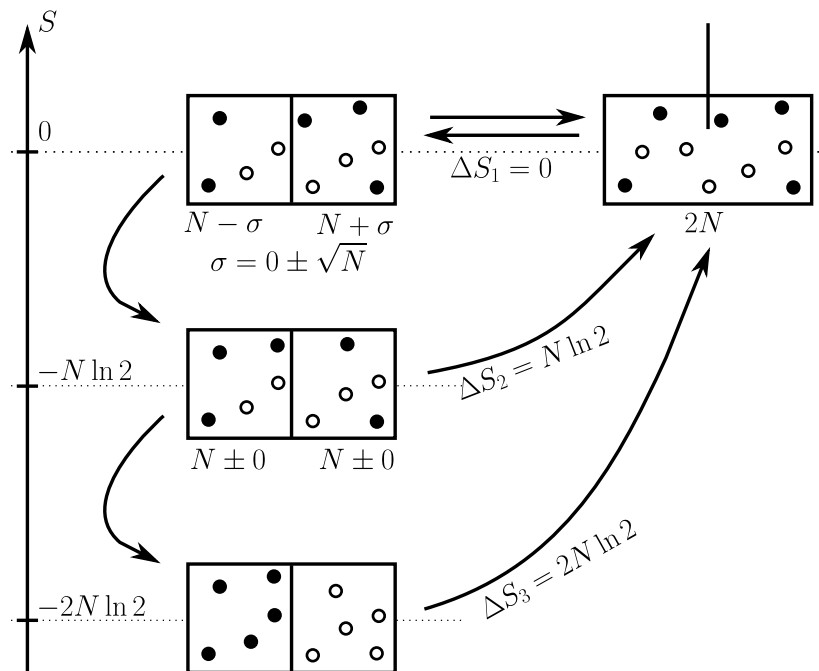

**Figure 3.** Mixing–unmixing cycle of a gas made of two species. The cycle depends on the information we had and do not want to lose. Top: information about the exact number of particles per compartment and traceability is lost at the end of the first cycle; therefore, further cycles leave it unchanged ($\Delta S_1 = 0$). Middle: retrieving exactly the same number of particles in each compartment has a minimal energy cost of $N \ln 2$ ($\Delta S_2 = N \ln 2$). Bottom: preserving particles' traceability has an additional minimal energy cost also equal to $N \ln 2$ ($\Delta S_3 = 2N \ln 2$).

Depending on our knowledge about the original state or depending on what we consider as being important about it, the mixing–unmixing cycle differs and the Shannon entropy of mixing does too. The latter can gradually take any value from 0 to $2N \ln 2$ by

steps of ln 2. The two Gibbs paradoxes are solved. Concerning the first Gibbs paradox of discontinuity, it makes sense only in the case where the dissimilarity of two volumes of gas is a continuous variable. Otherwise, one cannot speak about a discontinuous function but only about a discrete function. Two species, a and b, are either identical or different, there is no intermediate case. By mixing them, we can obtain all intermediate cases between pure-a and pure-b volumes. Then, we have shown that the entropy of mixing is continuous, so the first Gibbs paradox is actually falsidical. Pailluson [38] reaches the same result by considering polydisperse gases and allowing their dissimilarity to vary continuously. Concerning the second Gibbs paradox, introducing information shows that there is actually no contradiction between the calculation of Boltzmann (so that no correction is needed) and the result of thermodynamics. The two deal with different cases. The second Gibbs paradox is actually veridical.

### 3.2. Paradoxes Related to H-Function

Boltzmann was the first [4] to write a quantity defined at any time $t$, named H-function, which takes the form of an entropy. For that, he considers the phase space of one single particle ($\Gamma_1 \subset \mathbb{R}^6$) and the probability $p(\mathbf{x})$ to find a particles at $\mathbf{x} \in \Gamma_1$ at time $t$ (with a time scale supposed to be discretized like the phase space). The H-function is defined as

$$H(t) = \sum_{\mathbf{x} \in \Gamma_1} p(\mathbf{x}, t) \ln(1/p(\mathbf{x}, t)) \tag{24}$$

which is nothing other than Shannon's entropy of the particle density $p$.

Starting from the idea of Maxwell's kinetic theory of gases, that the motion of colliding rigid particles is governed by the equations of classical mechanics, but also that their large number allows statistical treatment, Boltzmann obtained an integro-differential equation (named the Boltzmann transport equation) for the time variation in the density $p$ of a dilute gas (for a reference book, see [50]). Boltzmann's equation allowed him to prove that, for a bounded phase space (an isolated system),

$$\frac{\mathrm{d}H}{\mathrm{d}t} \geq 0 \tag{25}$$

with the equality corresponding to the equilibrium state defined as stationary. Equation (25) is known as the H-theorem. *"Its proof is clever and beautiful, although not perfectly rigorous."* (Villani [51]); in fact, the proof in the general case is still in progress. But for physicists, the H-theorem is quite natural and can be viewed as another expression of the second law of thermodynamics [52] (concerning another distribution than that of microstates) plus a definition of equilibrium that warrants its stability. The second law states that the entropy of an isolated system cannot spontaneously decrease. So, even if classical thermodynamics (those of Clausius) say nothing about what exactly happens during a spontaneous process but only deal with the entropy before and after (the time variable is not present in classical thermodynamic equations), it is legitimate to say that entropy increases with time during a spontaneous process. For instance, consider a gas that is initially confined in a small box inside a larger room. Opening the box allows the gas to expand freely over an increasing volume ($\mathrm{d}H/\,\mathrm{d}t > 0$) until it uniformly occupies the entire room ($\mathrm{d}H/\,\mathrm{d}t = 0$) in a stationary state. In doing so, its entropy continuously increases.

The Boltzmann transport equation and H-theorem constitute the first attempt to demonstrate the macroscopic second law of thermodynamics from what happens at the microscopic scale. Against this attempt, two objections have been raised: the reversibility paradox and the recurrence paradox.

### 3.2.1. Loschmidt's Reversibility Paradox

This paradox was originally stated [53] in the form of a thought experiment. Consider the free expansion of a gas enclosed initially at time $t_0$ in a box placed in a larger room. Once the system is at equilibrium, after a certain time $\tau$, imagine that the direction of the

velocity of each particle is reversed, without changing its magnitude. The operation does not change the macroscopic properties of the gas, such as temperature or volume. So, neither heat nor work are provided to the system. But, once this has occurred, the gas particles go backward through the same sequence of collisions as the previous one. So, at time $2\tau$, its original microstate is restored. The gas is returned to the box in contradiction with the second law of thermodynamics and with the observation that this never happens.

The problem can be viewed in two different manners. First, to decide whether or not the process violates the second law, we must wonder how the operation of reversing the velocities of particles is possible and whether it can occur without energy expenditure (certainly not if the operation is physically performed with a device that obeys the second law). This problem, thus posed in term of an operating "demon", will be discussed in Section 3.3.2.

The other viewpoint is that this paradox basically raises the question of how, from time-symmetrical equations of motion (those of mechanics), it is possible to obtain time-asymmetrical results. The consensual answer [52,54,55] is that, within the ingredients that permit us to write the Boltzmann transport equation, the time asymmetry is already present in the form of the "hypothesis of molecular chaos": the velocities of two particles before their collision are fully uncorrelated but, of course, are fully correlated and determined by mechanics after the collision. Fundamentally, the Boltzmann transport equation (and thus the H-theorem) is obtained by moving the time asymmetry of the second law of thermodynamics from the macroscopic to the microscopic scale. The second law is phenomenological and comes from an inductive reasoning, which basically is a generalization of observations. By moving at the microscopic scale, it becomes a postulate or a hypothesis allowing us to build a deductive reasoning. This looks like circular reasoning, but actually for a theory, it is a progress in terms of economy of thought and potential unification of different areas of physics (for instance, the unification of thermodynamics and fluid mechanics).

However, for the purpose of this paper, two points are worth emphasizing. The first is that, in the spirit of the mechanical approach, the independence of probabilities for the velocities of pre-colliding particles results from the impossibility of reaching a sufficient accuracy of the initial conditions, i.e., it results from an incomplete knowledge (in this mechanistic conception, real stochasticity does not exist). The underlying conception of probabilities is therefore much closer to that of subjectivists than to that of objectivists (despite a frequentist ambition).

The second point is that, whatever its origin, that is to say, either incomplete knowledge (usual meaning of chaos) or true stochastic process, the molecular chaos results over time in a loss of correlation between microstates along the trajectory in the phase space. The system is no longer deterministic in the sense that the chain of causality, representing the trajectory of the system in the phase space, is broken. The volume-preserving dynamics (Equation (14)) and the ergodic hypothesis can still be postulated but they have lost their principal physical justification and the corresponding inductive reasoning has lost its strength and is much more hypothetical.

### 3.2.2. Poincaré–Zermelo Recurrence Paradox

Here comes in to play the Poincaré recurrence theorem. Consider a system with a bounded phase space $\Gamma$ that continuously undergoes over time a transformation $F$ that preserves the volume of any subset of $\Gamma$. Hamiltonian systems obey this condition according to Liouville's theorem, but here the condition is more general and can apply not only to deterministic but also to stochastic (purely random) systems like the Ehrenfest urn model [56]. Then, Poincaré shows that the system will recurrently pass to any point of $\Gamma$ already visited.

Going back to the example above of the free expansion of a gas from a small box into a larger room, yes the gas expands, but it is also expected to return to the box on its own without any demon. Although it would take a long time, it is not impossible, and not just once but recurrently.

Hence, the paradox stated by Zermelo (for a historical perspective, see [57]): How can such a recurrence be consistent with a continuously increasing H-function? How, also, can it be consistent with a stable state of equilibrium?

Different arguments have been put forward to resolve the Poincaré–Zermelo paradox. For instance, no system has a strictly bounded phase space; even the universe is expanding. Or, in the thermodynamical limit of an infinite number of particles, the time of recurrence is also infinite. The argument initiated by Boltzmann himself is that, for concrete thermodynamic systems with a very large number of particles, the calculated average time of recurrence is greater than the age of the universe. Practically, the gas never returns to the box on its own. So, everything is a question of a time window: *"The range of validity of Boltzmann's equation . . . is limited in time by phenomena such as the Poincaré recurrence"*(Villani [58]), but this limitation is never reached.

All these arguments are valid for resolving the paradox: they all amount to saying that, in practice, there is no recurrence. But it remains a problem. If there is no recurrence, how to conceive probabilities as frequencies? Or in the reverse manner, if there is recurrence, how to reconcile it with the second law (H-theorem) and with a stable equilibrium? The probabilist inference offered by information theory avoids this inconsistency.

*3.3. Demons*

Demons observe thermodynamic systems, acquire information about them and use it to act on them. In doing so they can possibly produce energy. Where does this energy come from? In fact, energy is an abstraction only defined by a conservation principle [59]. So, if something is missing in an energy balance, it means that we have discovered a new form of energy. Demons, on their own, demonstrate the link between information and energy. The same idea can be expressed in another manner: *"In so far as the Demon is a thermodynamic system already governed by the Second Law, no further supposition about information and entropy is needed to save the Second Law."* Earman and Norton [60]. In other words, the very definition of a principle is that everything conforms to it; by definition, a principle is inviolable.

This is an application of pure logic with which I completely agree. However, given the great expenditure of gray matter devoted to this question, such an answer cannot suffice. *"How does it happen that there are people who do not understand mathematics? If the science invokes only the rules of logic, those accepted by all well-formed minds, if its evidence is founded on principles that are common to all men, and that none but a madman would attempt to deny, how does it happen that there are so many people who are entirely impervious to it?"* (Poincaré [61] p. 46). In fact, demons raise paradoxes that exist, and continue to exist even after they have been "resolved", by the mere fact that they have been stated. So, we cannot shrug them off only by pure logic: we need more. Thus, in what follows, we do not use the shortcut given in the preamble and rather examine whether what we know from the encoding problem is sufficient for understanding how demons can operate in accordance with the second law. No further supposition about information and entropy is needed to save the second law [60], but we suppose that we already have information theory at our disposal to solve the inconsistencies raised in the previous section. So, here, we just want to check its consistency with demons.

With the encoding problem, the quantity of information needed to represent a system, or equivalently the uncertainty about its state, is identified with its entropy, thus linked to energy. In particular, acquiring information, i.e., reducing uncertainty, requires an energy expenditure (Equations (11) and (12)). This acquired information is similar to potential energy: it is stored and could be used in return. Increasing the potential energy requires work, but to use it in return, something else is needed: know-how. Otherwise, the potential information energy simply dissipates at the end of a cycle, which is when the information is outdated. In short, it is not the acquisition of information that directly has an effect on the system, it is the acquisition plus the action that depends on it. A misunderstanding of this point is at the origin of ill-founded criticisms of information theory (e.g., [62]).

Demons are supposed to know how, but the realization requires a physical implementation, not only of the action itself but also of all the information processing chain. By the physical implementation of information processing, I mean a black box including everything necessary for measurement, storage, transmission, eventual erasure, etc., that necessarily fall under the second law of thermodynamics. It is this physical implementation that is responsible for the minimum energy expenditure for demons to operate.

From the second law of thermodynamics expressed in terms of information (Equations (11) and (12)), the acquisition of 1 bit of information has a minimum energy cost equal to $T \ln 2$. Therefore, to check if demons work in accordance with the second law, it is enough to check if the quantity of information necessary for their action is consistent with the energy that can be obtained in return.

### 3.3.1. Maxwell's Demon

The family of thermodynamical demons [63] was born with the temperature demon of Maxwell [64]. Imagine a gas in an insulating container separated in two compartments *A* and *B* by a thermally insulating wall in which there is a small hole. A demon "*can see the individual particles, opens and c1oses this hole, so as to allow only the swifter particles to pass from A to B, and only the slower ones to pass from B to A. He will thus, without expenditure of work, raise the temperature of B and lower that of A in contradiction to the second law of thermodynamics.*" ([64] p. 308). The temperature difference between the two compartments can eventually be used for running a thermodynamic cycle and producing work.

A simplified version is the pressure demon: particles whatever their speed can only pass from *A* to *B*. This results in a pressure difference between the two compartments, which can be used for producing mechanical work. Alternatively, in this simplified version, the demon can be replaced by a concrete device, either by a one-way valve, as proposed by Smoluchowski [63], by a ratchet–pawl mechanism [65], or by an electric diode and the gas particles by electrons [19,66]. Then, if the two compartments communicate by an additional channel, the device is expected to rectify thermal fluctuations and produce a net current of particles, which can deliver useful energy. For the last two concrete devices, it has been experimentally shown that they can work provided that the rectifier (ratchet–pawl or diode) is cooled at a lower temperature than the rest of the system [67,68] in return for the entropy decrease. Demons work in the same way once physically implemented.

The quantitative verification of the correspondence between, on the one hand, the information necessary for the demon and, on the other hand, the energy that can be obtained in return is simplified with the device proposed by Szilard [69]. It is composed of a single particle in a volume $2V$ (see Figure 4). The demon does not care about the velocity of the particle but only the compartment it is in. This information is encoded with only 1 bit, which costs at least $T \ln 2$ (Equation (12)). In the opposite compartment, the demon installs a piston, which encloses the particle in a volume $V$. The system can return to the original state by an isothermal expansion that provides to the surroundings a work equal to, at best, $T \ln 2$. The overall cycle is consistent with thermodynamics.

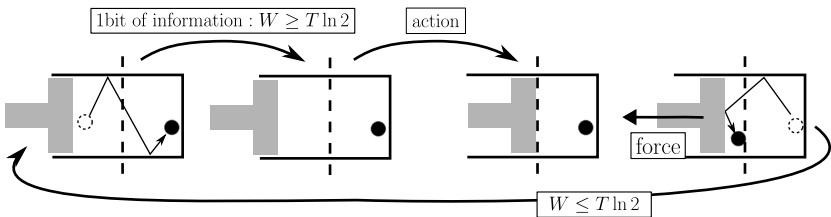

**Figure 4.** Szilard demon detects when the particle is in the suitable compartment and then installs a piston, allowing the device to subsequently produce work.

### 3.3.2. Loschmidt's Demon

Let us return to the Loschmidt's paradox of reversibility (Section 3.2.1), this time with an operating demon actually able to reverse the velocities of particles. We wonder whether

the quantity of information (in terms of the minimum number of bits to encode it) necessary for the demon to operate is in agreement with the mechanical work that the gas could produce with a subsequent isothermal expansion.

Let $V_1$ and $V_2$ denote, respectively, the volume of the initial box and that of the room in which the gas was expanded, and $\lambda$ such as $V_2/V_1 = 2^\lambda$. So, the mechanical work per particle provided by an expansion is equal to, at best, $\lambda T \ln 2$, and for $N$ particles,

$$W \leq N\lambda T \ln 2 \tag{26}$$

For the demon to reverse the velocity of one particle in volume $V_2$, it must intercept the trajectory of the particle with an elastic wall (a mirror) having the correct direction (perpendicular to the trajectory), the correct orientation ($+$ or $-$) and the correct position (that of the particle) [70]. All the necessary information resides in the recording of the corresponding microstate of the particle. From $V_1$ to $V_2$, the number of bits required for encoding the velocity of one particle remains unchanged, but encoding its position requires $\lambda$ extra bits (the cardinality of the phase space of one particle increases by a factor $V_2/V_1$). For $N$ particles, $N\lambda$ extra bits are needed. From Equation (12), their acquisition costs at least $N\lambda T \ln 2$, in agreement with the work that can be obtained in return and with the second law of thermodynamics.

### 3.3.3. Landauer's "Principle"

Equation (12), obtained in the sole framework of Shannon's information theory (plus the second law of thermodynamics), strongly resembles another one known as Landauer's "principle" [71–74] (which also uses the second law but is free of Shannon's information theory and of the encoding problem). Clarification is therefore necessary to avoid confusion.

Landauer considers the physical implementation of a logical bit in the form of one-to-one mapping between the two logical states (0 and 1) and two thermodynamical states materialized, for instance, by a particle in a bistable potential. In this framework, it was shown that the irreversible logical operation ERASE (or RESET TO 0) of the bit can be split into two steps:

1.  Put the bit into an undetermined state by flattening the potential;
2.  Set the bit to 0 by applying a bias and then raise back the energy barrier.

The point is that during the first step, the probability distribution of the particle undergoes a leakage comparable to the irreversible adiabatic free expansion of a gas (by a factor 2). So, neither heat nor work are exchanged with the surroundings. Comparatively, the second step can be quasistatic. Suppose that the initial state was 0, the second step closes a thermodynamic cycle. So, to be in agreement with the second law, it must have an energy cost at least equal to $T \ln 2$. It follows that the total energy cost of the operation ERASE (of the cycle) is

$$W_{\text{erase 1 bit}} \geq T \ln 2 \tag{27}$$

This result is known as the Landauer's "principle", despite the fact that it cannot be a general principle (hence the quotes) but only applies to this particular physical implementation. Actually, to avoid any leakage from one potential hole to its neighbor, it is enough to design a physical implementation based on a two-to-one mapping between logic and thermodynamic states [75]. With only one potential hole, there is no leak.

Equation (12) has general validity and concerns the acquisition of a data bit, whatever the way in which it was carried out and including all the steps necessary for this acquisition. Equation (27) concerns the erasure of a data bit with a particular physical implementation consisting of a bistable potential and results from the thermodynamics of this particular case. The difference between "acquisition" and "erasure" should be clear enough to avoid confusion. But we can conceive certain particular data acquisition procedures (in particular that envisaged by Landauer and Bennet [72,74] in which they propose to replace the thermostatistic demons) that require erasing the bit before writing a new value there. In this case, Equations (12) and (27) lead to the same result. Hence, a possible confusion.

The generic black box of the demons based on Equation (12), which dissipates $T \ln 2$ per bit, includes everything necessary for measurement, storage, transmission, eventual erasure, etc. Different physical implementations correspond to different places where dissipation could occur. There is absolutely no clue that allows us to suggest that this place is universal. Brillouin analyzed the physical limits for an observation through numerous examples of measurement procedures that could be implemented [19]. He showed that, in all cases, the energy expenditure needed to reach a given accuracy and the corresponding decrease in entropy that this information would allow are consistent with the second law of thermodynamics.

But, currently, the most popular physical implementation is that of Landauer. The functioning of Landauer's black box is such that the measurement is free and only its recording in the form of bit values causes energy dissipation. For 1 bit of data, this functioning is as follows (see Figure 5):

1.  The bit is materialized with a bistable potential. Erasing the bit dissipates at least $T \ln 2$ (Equation (27)).
2.  The recording procedure requires the bit to be erased before new data are recorded.

This functioning is a doubly special case: a special case of bit implementation and a special case of recording procedure. Based on the second law, it is obviously in full agreement with Equation (12). So, if it is claimed that a solution using Landauer's "principle" is found for the paradoxes introduced by demons, then the same solution is valid using the sole framework of Shannon's information theory and Equation (12). But, this time, in a more direct way with general validity.

Landauer's "principle" is presented as the key point for definitively resolving the paradoxes caused by demons and, therefore, to definitively link information to energy [70,76–82]. In addition to the previous objection that it is not a general principle, let me focus on the second point of the functioning of Landauer's black box for demons.

When I was a teenager, I had a boombox to record my favorite music. With this device, it was possible to fully erase an already-used cassette to start with an almost blank tape (a standard state). The idea behind this was that if you leave a blank between two pieces of music, you will not hear the old music when listening to the new. But erasing the cassette was not mandatory. The cassette could be directly overwritten, for example, if a long concert was recorded. In this case, the silence between two pieces (as silences within a given piece) is not a blank (an absence of message) but a message in itself. In other words, it is possible to record and process data without having to erase anything (whether the data are digitized or not does not change anything). The injunction to avoid overwriting (and thus the need of erasing), which we find in the recent literature (e.g., [70]), is unfounded.

Actually, ERASE and OVERWRITE are irreversible logical operations. The logical irreversibility is defined by Landauer himself: *"We shall call a device [an operation] logically irreversible if the output of a device does not uniquely define the inputs."* [71]. It is a property of the initial and final states of the bit. For instance, a logical operation from *A* to *B* is logically irreversible if, once in *B*, the information of where the system was initially has been lost, so it is not possible to return to the starting point, *A*. On the contrary, a transformation is thermodynamically irreversible if it is not possible to return to *A* using the same path backwards. Thermodynamical irreversibility is a path property. Therefore, it is not surprising that a logical irreversible operation can be achieved by using a reversible thermodynamic process.

Generic implementation

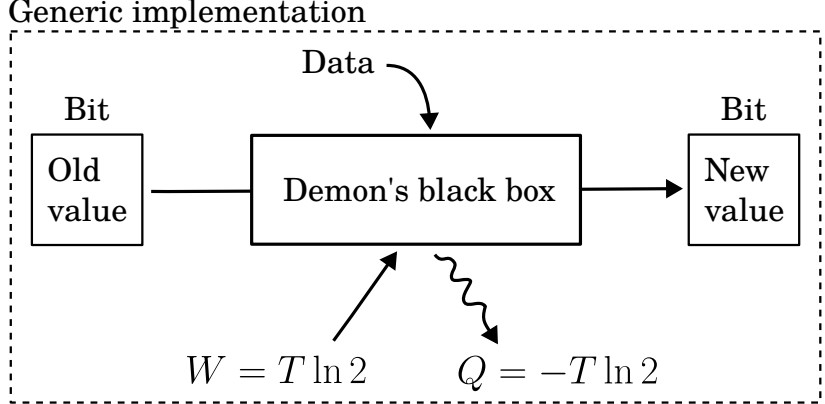

Landauer implementation

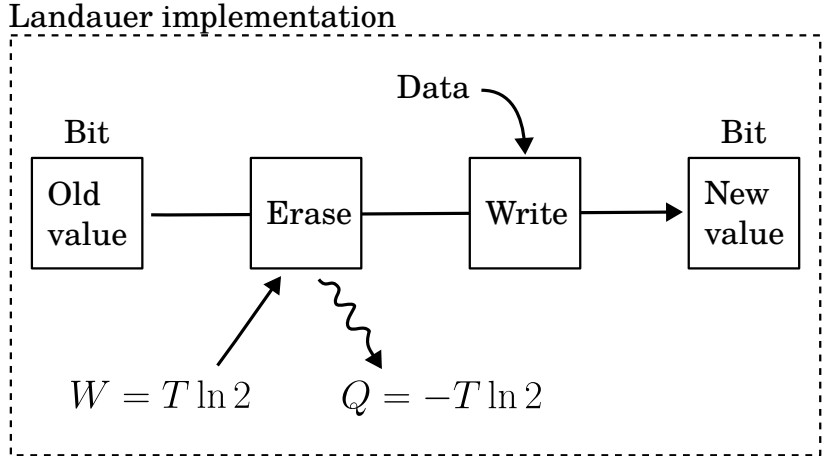

**Figure 5.** Generic implementation (Equation (12)) versus Landauer implementation (Equation (27)) of a demon.

## 4. About Subjectivity in Physics

The approach of information theory to statistical mechanics avoids all inconsistencies of the alternative frequentist position. But we are forced to note an opposition to this idea. This opposition goes beyond science and is epistemological. The subjectivity introduced by information theory is the sticking point. Among many quotes, I note these which are particularly clear in this regard:

> *"The Jaynes approach [that of maximum entropy principle] is associated with a philosophical position in which statistical mechanics is regarded as a form of statistical inference rather than as a description of objective physical reality."* (Penrose [2])

> *"A number of scientists and information theorists have maintained that entropy is a subjective concept and is a measure of human ignorance. Such a view, if it is valid, would create some profound philosophical problems and would tend to undermine the objectivity of the scientific enterprise."* (Denbigh and Denbigh [17])

> *"This [the maximum entropy principle] is an approach which is mathematically faultless, however, you must be prepared to accept the anthropomorphic nature of entropy."* (Lavis [83])

The aim of this section is to show that the type of subjectivity introduced into physics by information theory is in reality not new at all. It is in line with an ancient development of a general conception of what science is, which I propose to clarify.

*4.1. Representationalism*

The first subjective character introduced by information theory lies in the problem of coding a representation of the system that allows its behavior to be reproduced.

This representation depends on our knowledge, in the sense that it depends on the state of the art of the devices used to measure and probe the parameters needed for this representation. It also depends on the parameters that we consider relevant for this representation. Consider, for example, the unavoidable impurities in any chemical substance. A correct representation of the system must take them into account. But, below a given threshold, which depends on our tools of measurement, impurities are no longer detectable and cannot be part of our representation of the system. But we can suppose them to be still present in an objective being of the system. Impurities are present in the real system but not in our representation of it. Additionally, impurities can be isotopes. What about the representation of the system before their discovery? [46]. Also, in the thermodynamics of motors, pumps etc., most of the time we do not care about isotopes (actually, we do not care about atoms either; before their discovery, thermodynamic engines already worked very well), so their presence or absence is useless information for the representation of the system (and does not need to be encoded). In this sense, information is subjective. It depends on the state of knowledge of the observer or which level of detail we (collectively) consider as relevant to describe a system. But it does not depend on the personality of the observer; it is not a personal element [15].

The subjectivity of entropy was already acknowledged by Maxwell (*"The idea of dissipation of energy depends on the extent of our knowledge"* [18]) and Gibbs (*"It is to states of systems thus incompletely defined that the problems of thermodynamics relate."* [36]). This is believed to contrast with other physical quantities considered objective properties. Actually, information theory does nothing other than explicitly introduce representationalism (also named indirect realism) into these problems.

The basic idea of indirect realism is that our only access to reality is that provided by our senses (in a broad sense, that includes all laboratory instruments). Following Einstein, *"all knowledge about reality begins with experience and terminates in it"* [25]. If "experience" is understood as a conscious event that passes through our senses, it follows that the concern of science is not the reality but the representation our senses give us of it. Fifty years before Shannon, Mach wrote, *"The law always contains less than the fact itself, because it does not reproduce the fact as a whole but only in that aspect of it which is important for us, the rest being either intentionally or from necessity omitted."* [84]. He was not talking about entropy or thermodynamics, he was talking about the laws of physics in general.

Information theory formalizes this idea, which would otherwise remain unclear and implicit. Entropy itself is a very objective property and is well defined mathematically. But it is an objective property of a subjective representation of reality. In this, according to indirect realism, entropy does not differ from other physical quantities. The above argument should be able to answer the question asked by some: *"Thermodynamic entropy is not different, in regard to its status of objectivity, from physical properties in general. How came it then that so many scientists have held, and still hold, the opposite opinion?"* (Denbigh and Denbigh [17] p. 18).

Regarding all physical quantities other than entropy, from a purely scientific point of view, the difference between direct and indirect realism is just a question of vocabulary. For direct realists, science is directly about the real world. For indirect realists, we do not have access to the real world but only a representation of it, so only this representation is the subject matter of science. This difference can be ignored in the daily practice of science. Rename "representation of reality" to "reality" and both realisms are talking about the same thing. In both frameworks, theories concern the same object and their experimental examinations are equally achieved from our interactions with this object. The difference is only epistemological. Poincaré wrote: *"Does the harmony [the laws of nature] the human intelligence thinks it discovers in nature exist outside of this intelligence? No, beyond doubt, a reality completely independent of the mind which conceives it, sees or feels it, is an impossibility.*

*A world as exterior as that, even if it existed, would for us be forever inaccessible. But what we call objective reality is, in the last analysis, what is common to many thinking beings, and could be common to all; this common part can only be the harmony expressed by mathematical laws. It is this harmony then which is the sole objective reality, the only truth we can attain."* ([85] p. 15).

When it comes to entropy as seen by information theory, what becomes troubling is that the difference between the direct and indirect realism views can no longer be ignored, even in science. This is the only point on which entropy is so special compared to all other physical quantities. Shannon's entropy quantifies the complexity of the representation itself (in this case, the quantity of information is a particular case of complexity [13]). In doing so, it makes the notion of representation crucial and explicit.

*4.2. Induction*

Science is linked to knowledge, understood as a set of statements recognized as true. According to logical empiricism (or logical positivism), we have two possible sources of knowledge, each linked to a type of reasoning that assert that a statement is true: the first is purely logical (deductive reasoning) and the second empirical (inductive reasoning). Here, we will not enter into the debate on the justification of deduction, that is to say, on the origin of the elementary rules of natural logic (which can possibly be empirical), we will focus only on induction, but we will need deduction for comparison.

Induction is unavoidable and omnipresent in natural sciences: generalization, interpolation, regression analysis, analogy, etc., are all inductions based on known experimental facts. *"Without generalisation, prediction is impossible"* (Poincaré [86]). In fact, induction is the reasoning that allows us, from our current knowledge, to predict new observations or answer new questions. At the basis of phenomenological laws, but also theoretical hypothesis, postulates or principles, there is always inductive reasoning, at least implicit.

However, if the truth of a deductive statement can be proved and verified (provided the premises and the logical rules are right), the verification of an inductive statement can never be definitively achieved because this would imply a infinite non-countable set of experimental facts. The truth of a deduction is certain; that of an induction is at best probable. *"By generalization, every fact observed enables us to predict a large number of others; only, we ought not to forget that the first alone is certain, and that all the others are merely probable"* (Poincaré [86]). If known experimental facts make it possible to base an inductive reasoning, new or upcoming ones can only either confirm or refute it, but never definitely prove it. Inductions are by essence provisional and likely to be updated or replaced by better ones as progress is made.

If an inductive statement can never be verified (proven to be true), how can we differentiate between a well-founded scientific claim and another that is ill founded and irrational? How can we make a hierarchy between different reasonings? Which is the best? This is known as the problem of induction (for a recent book on this topic, see [87]).

A first piece of answer was provided by the falsificationism of Popper [88]. Since verifiability cannot be required for induction, Popper instead suggests replacing it with falsifiability. A valid inductive reasoning must be falsifiable (or refutable): it must be able to be confronted with the experiments. This is the first condition. If it is met, an induction remains "true" until proven otherwise. The requirement of falsifiability of an induction entails another: that of not being tacit or hidden, but explicit. Otherwise, we make the reasonings without any chance of attempting to refute them [86]. But this is still not enough to establish a hierarchy of inductive reasoning.

Confirmation or refutation of inductive reasoning passes though experiments. At first glance, the refutation seems clear-cut, while the confirmation seems gradual (incremental) as more and more facts consistent with an induction reinforce it. However, both are conditioned on the validity of experimental results, themselves conditioned on confidence intervals (errors bars). This automatically introduces a link between the notion of the "degree of confirmation" or "degree of belief" and that of the "probability of truth" of an induction [89]. Hence, the claim that all inductive reasoning in science falls under the same

universal pattern as that of probabilistic inferences. The best is the most probable according to our present knowledge, that is to say, the one that has the highest prior probability of being true. This is the essence of Bayesianism [90] (named after Bayes and his theorem about the probability of an event conditioned on prior knowledge) and its derivatives in spirit, among which the maximum entropy inference can be classified.

Not everyone agrees with the existence of such a universal pattern for induction. For example, Norton [91] introduces a material theory of induction that professes that the logic of induction is determined by facts specific to each case and which cannot always be expressed in terms of probability. To which it has been opposed [92] that as soon as the confirmation procedure (and then the updating of the induction) involves data and measurements, probabilities come into play.

Whether maximum entropy inference is a starting point to produce a first prior probability distribution necessary for Bayesian updating or whether it is a generalization of Bayesian inference, unless it is the other way around [93], is beyond the scope of this paper. The main point is the universal aspect of all inductive reasoning, that of being ultimately probabilistic, that of involving prior knowledge and prior probabilities and that of being subjective (in the sense given to this word in this paper).

Although scientists are aware of the problem of induction and adopt probabilistic inductive reasoning for their daily practice (personally, I do not know any scientist who would prefer to work on the option they believe has the lowest probability of success), this practice is not necessarily conscious and the problem of induction is often (temporarily) forgotten or denied. Below are some quotes from the recent literature of interest here: *"Experimental verification of Landauer's principle linking information and thermodynamics."* (Bérut et al. [76]); *"Information and thermodynamics: experimental verification of Landauer's Erasure principle."* (Bérut et al. [77]); *"We experimentally demonstrate a quantum version of the Landauer principle."* (Yan et al. [79]); *"Landauer's principle has been recently verified experimentally"* (Binder [70]), etc. I cannot imagine that these authors ignore or disagree with the impossibility of verifying an induction. Instead, I interpret these quotes as language facilities that are not innocent but reveal a reluctance to inductions. Physicists prefer deductions, proofs and definitive verifications; all things expected of hard sciences.

The problem with information theory is that here, again, as for representationalism, everything is explicit: we cannot feign ignorance of our complete dependence on subjective probabilities in natural sciences.

## 5. Conclusions

The subjectivity of information theory as it has been defined in this paper, that is to say, something that is not personal but simply refers to the role played by our knowledge, allows us to resolve the inconsistencies present in thermostatistics from the start. At the same time, it is this subjectivity that worries some for epistemological rather than scientific reasons.

The role played by subjectivity should not be so surprising, at least in this area. Thermodynamics from the beginning refers to anthropocentric concepts and vocabulary, such as energy grades, useful energy, energy cost, work, dissipation... In addition, thermodynamics is a science of the macroscopic scale. This term itself is anthropocentric, since macroscopic only designates our human scale. Indeed, in practice, the role that a certain subjectivity can play is admitted in science. But we are so steeped in positivism and with the ambition to be objective that when subjectivity becomes too explicit it becomes annoying. In fact, science is a human construct. The "Laws of Nature" do not come from nature, they come from us. The mere fact that these laws are provisional and subject to being continually replaced (updated) by better ones as science and our knowledge progress proves this.

Finally, there is another source of reluctance towards information theory that can be perceived in light of certain recent publications. It was not mentioned in this article but probably deserves special attention. It is also linked to the ambition of objectivity, but not in the same way that the refusal of representationalism and Bayesianism was. This is due to a particular meaning given to the word "physical", understood as "materialized", as

opposed to virtual or non-tangible. There is nothing more "objective" than matter. After Landauer ("Information is physical" [72]), many authors interpreted his principle as the missing element they were waiting for to materialize information. Probably the most recent development of this idea can be found in the "mass-energy-information equivalence principle" [94,95]. In short, information is energy, and energy has a mass equivalent in special relativity ($E = mc^2$); therefore, information has a mass. For example, the author proposes measuring the mass of a hard drive before and after erasing 1 TB of data. The idea behind it is that information is a kind of potential energy, which can be used for instance by a demon to act on a system and produce work. What is the status of potential energy in special relativity? [96,97]. Actually, potential energy is not energy, it is something that is potentially energy. It is actually stored in the form of rest mass [98], just like the mass defect in nuclear physics. Hence, the idea behind the "mass-information equivalence principle". Potential energy is a concept, just like entropy and information are. But there is a major difference: if we wonder about the mass equivalence of $T\Delta S$ when a body undergoes a process, we implicitly consider a constant temperature and therefore a constant internal energy to which no variation in rest mass can be associated as being localized in the body considered, but more likely in its surroundings. In this sense, entropy and information remain even more elusive concepts than that of potential energy. But developing this point deserves another article [99].

**Funding:** This research received no external funding.

**Data Availability Statement:** The author declares that there are no data to share attached to this paper.

**Conflicts of Interest:** The author declares no conflicts of interest.

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
