# Peer review of "Thermostatistics, Information, Subjectivity: Why Is This Association So Disturbing?"

_mathematics, doi:10.3390/math12101498_

Round 1

Reviewer 1 Report

Comments and Suggestions for Authors

The manuscript introduces the subjectivity of information theory to resolve the inconsistencies in thermostatistics. The objective and subjective conception of entropy is discussed and the inconsistencies between the two approaches can be eliminated with the information theory. The paper is well-structured and well-written. The author adequately provides the theoretical aspects related to the topic. However, the author should clarify the following issues:

1. Since the manuscript is just a synthesis of previous research, what is the "new" aspect here? The author must clarify it.

2. A lot of theories are presented in the manuscript. However, the results supporting the conclusions are not clearly seen. The author should provide a specific example of how the information theory resolves inconsistencies in thermostatistics.

Author Response

1) The "new" aspect is precisely in the revue, the didactic point of view that is adopted and the clarification of the literature. It is true that probably everything (and its contrary) has been already written on this topic, so that the main goal of the paper is an attempt of clarification and to show that the main problem is epistemological (rather than scientific). I am glad to note that in view of the repports received the target is reached.
An advise for the readers about the nature of the paper has been added in the introduction (line 101-102).

2) The best example is likely in the proposed solution for the Gibbs paradoxes of mixing two gases (section 3.1). This example is detailled over 4 pages in the paper. Incidently, note that the solution here proposed is not that commonplace...
Further examples would probably desequilibrate the paper that mainly intends to put the debate in section 4 at an epistemological level (as it is clearly announced in the abstract and in the introduction).

Reviewer 2 Report

Comments and Suggestions for Authors

The paper is a discussion of thermodynamics ("thermostatistics") from the perspective of information theory. It is well written and and reviews the topic extensively without the polemic attitude that is often characteristic of the subjective/frequentist debates. I have a couple comments that I hope the author would address:

With respect to the Gibbs paradox, one of the resolutions discussed by the author is based on the indistinguishability of particles, which, the author adds citing Huang (ref 38), ``comes from quantum mechanics.'' Uffink on the other hand (ref 53), remarks that many others, including van Kampen (ref 44) dispute that quantum indistinguishability is the crux of the paradox. From an operational perspective particles can be distinguished if they can be separated on the basis of different physical properties. The case of isotopes, also discussed in section 3.1.3,  is an example of nearly but not quite identical particles, whose separation is possible but difficult. That is, the separation work of mixture is a continuous function of the difference in the physical properties that distinguish the species. The example further bolsters the ``subjective'' view of entropy. Suppose that we discover in the future new properties that identify subspecies of oxygen, similar to isotopes, leading to separation of these subspecies. Then the entropy of mixing of oxygen in the future will be larger than we think today on account of the mixing of these subspecies. Entropy, in other words, reflects our current state of knowledge about the system at hand. 

I am not convinced that the solution of the Gibbs paradox in section 3.1.3 falsifies the paradox of discontinuity, which is tied to the discontinuity of species a and b. What it does show is that entropy is related to the information we wish to retain in the course of a process. 

Overall I find this a very nice work and recommend publication. 

Author Response

Concerning your first comment about (in)distinguishability of particles, my point is precisely to avoid this debate, mainly in order to avoid a polemic attitude. If this kind of debate were to succeed, it would have already happened. I avoid this endless debate by noticing that: either identical classical particles are indistinguishable for some reasons, but clearly these reasons depend on our knowledge (some particles which are not distinguishable today may become so in the future); or identical classical particles are always distinguishable (traceable) but then in the 2nd Gibbs paradox there is a loss of ergodicity when the two volumes disjoin. In both cases, the subjectivism is introduced. If you agree, I would much prefer to leave the text unchanged in this regard.

Concerning the 1st Gibbs paradox of discontinuity. It makes sense only in case the dissimilarity of two volumes of gas is a continuous variable. Otherwise, one cannot speak about a discontinuous function, but only about a discrete function. Two species a and b, are either identical or different, there is no intermediate case. By mixing them, we can obtain all intermediate cases between pure-a and pure-b volumes. In this case, we show that the entropy of mixing is continuous, so that the 1st Gibbs paradoxe is falsidical. Thanks to your remark, a paragraph has been added in the revised version (line 638-648), with a new reference given by reviewer 3.

Reviewer 3 Report

Comments and Suggestions for Authors

see attachment

Author Response

Submission to "Mathematics" is not my sole responsibility. It is the result of a long journey until an editor feels the paper suitable for peer-review.

Thank you for the reference that is fully relevent for my paper. It has been added to the revised version. It is now referred to this paper line 534 and line 644 (in a new paragraph about this topic).

Concerning your last comment, I am afraid that accounting for this remark would lead me too far from my purpose. My intention was not to be exhaustive, which would have been presumptuous in any case.